# Changes in Lake Sturgeon Gut Microbiomes Relative to Founding Origin and in Response to Chemotherapeutant Treatments

**DOI:** 10.3390/microorganisms10051005

**Published:** 2022-05-10

**Authors:** Shairah Abdul Razak, John M. Bauman, Terence L. Marsh, Kim T. Scribner

**Affiliations:** 1Department of Fisheries & Wildlife, Michigan State University, East Lansing, MI 48824, USA; shairah@ukm.edu.my; 2Department of Applied Physics, Faculty of Science & Technology, Universiti Kebangsaan Malaysia, Bangi 43600, Malaysia; 3Michigan Department of Natural Resources Fisheries Division, Escanaba Customer Service Center, Gladstone, MI 49837, USA; baumanj@michigan.gov; 4Department of Microbiology and Molecular Genetics, Michigan State University, East Lansing, MI 48824, USA; marsht@msu.edu; 5Department of Integrative Biology, Michigan State University, East Lansing, MI 48824, USA

**Keywords:** chemotherapeutants, environmental variation, founder effects, gut microbiome, lake sturgeon

## Abstract

Antibiotics, drugs, and chemicals (collectively referred to as chemotherapeutants) are widely embraced in fish aquaculture as important tools to control or prevent disease outbreaks. Potential negative effects include changes in microbial community composition and diversity during early life stages, which can reverse the beneficial roles of gut microbiota for the maintenance of host physiological processes and homeostatic regulation. We characterized the gut microbial community composition and diversity of an ecologically and economically important fish species, the lake sturgeon (*Acipenser fulvescens*), during the early larval period in response to weekly treatments using chemotherapeutants commonly used in aquaculture (chloramine-T, hydrogen peroxide, and NaCl_2_ followed by hydrogen peroxide) relative to untreated controls. The effects of founding microbial community origin (wild stream vs. hatchery water) were also evaluated. Gut communities were quantified using massively parallel next generation sequencing based on the V4 region of the 16S rRNA gene. Members of the phylum Firmicutes (principally unclassified *Clostridiales* and *Clostridium_sensu_stricto*) and Proteobacteria were the dominant taxa in all gut samples regardless of treatment. The egg incubation environment (origin) and its interaction with chemotherapeutant treatment were significantly associated with indices of microbial taxonomic diversity. We observed large variation in the beta diversity of lake sturgeon gut microbiota between larvae from eggs incubated in hatchery and wild (stream) origins based on nonmetric dimensional scaling (NMDS). Permutational ANOVA indicated the effects of chemotherapeutic treatments on gut microbial community composition were dependent on the initial source of the founding microbial community. Influences of microbiota colonization during early ontogenetic stages and the resilience of gut microbiota to topical chemotherapeutic treatments are discussed.

## 1. Introduction

Developing therapeutic regimes that limit stress-induced microbial infection or that reduces the occurrence of high mortality events in aquaculture is essential to successful fish production [1,2,3]. In aquaculture systems, stress in fish increases as a result of unfavorable rearing conditions (e.g., water quality, water source) or common production practices (e.g., handling, disease treatment), and interferes with physiological processes that aid in the defense against pathogens [3,4]. In response to a growing need for approved therapeutic regimes, fish culture managers have experimented with a variety of external (topical) disinfectant treatment strategies (hereafter referred to as “chemotherapeutants”). Indirect effects of these compounds, for example, associated with changes in gut microbial community composition and diversity, have not been rigorously evaluated in fishes [5]. However, disruptive effects of antimicrobial compounds on gut microbial communities are widely recognized in humans [6,7]. Given the propensity of larval fishes to internalize water and associated microbial communities via ingestion or respiration [5], one can postulate similar disruption to the gut microbiome of fish.

Common aquaculture treatment strategies include the use of chemotherapeutants (1) to treat infected fish as a function of visual detection of disease or in response to high mortality events, or (2) to administer regimented chemotherapeutant prophylactic treatments to reduce stress and prevent incidences of high mortality associated with pathogen infection [8]. Chemotherapeutant prophylactics used to reduce stress and prevent most prevalent disease-causing bacteria among cold-, cool-, and warm-water fish include chloramine-t (CT), hydrogen peroxide (H_2_O_2_), and sodium chloride (NaCl_2_) [8]. CT is an external disinfectant found to effectively treat fish with or by prophylaxis to prevent external bacterial infections [9,10], particularly those associated with flavobacteriosis [8]. Similarly, hydrogen peroxide is an oxidative external disinfectant that has been used in aquaculture since the 1930s [11], and has been shown to reduce or eliminate infections, improving survival across multiple species at multiple life periods [12,13,14,15]. For example, H_2_O_2_ has been used to control mortality associated with finfish egg Saprolegniosis, as well as mortality of larval and juvenile fish infected with external pathogens, such as *Flavobacterium* [8]. NaCl_2_ is one of the most commonly used chemotherapeutants for the control and treatment of external pathogens [16,17] as well as for osmoregulatory aid [8,18,19]. In addition, NaCl_2_ use is believed to be associated with the ‘shedding’ of the mucosal layers, which exposes potential pathogens to treatment [20]. The toxicity and effectiveness of chemotherapeutants utilized in aquaculture differs by fish species, treatment regime, treatment concentration, as well as the life period during which treatments are administered [16,21,22,23]. Given that approved chemotherapeutants were initially and most commonly assessed using salmonids, and largely associated with external infections [8,9,15,16], further research is needed to evaluate the applicability of common chemotherapeutants when internalized and for other fish species, including those of conservation concern, such as lake sturgeon (*Acipenser fulvescens*).

Community ecological theory (e.g., [24]) can play an important role in studies of microbial communities and aquatic animal health. Theoretical and empirical studies emphasize the effects of processes associated with patterns in diversity, abundance, and species composition. One established theory in community ecology involves drift or neutral stochasticity on random compositional variation associated with initial colonization [25,26,27]. Other processes associated with community compositional changes involve response to disturbance [28,29]. Disturbance can be defined as a “single disruptive event or set of events that significantly changes ecological community structure and function” [28,30]. Some microbial communities might experience irreversible changes in taxonomic composition and function, for example, certain populations may be extirpated. Other communities may be resilient, where compositional changes are transitory, and community composition and diversity returns to pre-disturbance levels. Due to high functional redundancy in microbial communities [31], changes in community composition may also occur due to changes to and/or loss of minor populations. However, there may not be appreciable change to community function as roles of newly added constituents maintain the role(s) of original community members [30,32,33,34].

Widespread use of chemicals, drugs, and antibiotics is an example of a disturbance to microbiota, and is a rising concern in aquaculture [35,36,37]. With recent expansion and rapid growth in demand for aquaculture products in conservation and food production [38], chemical and antibiotic applications are increasingly used in aquaculture to control pathogens [39,40]. While short-term benefits are often realized, there is potential for damaging impacts of these practices, including disruption of co-adapted microbial communities. Further, large amounts of chemotherapeutants are passed into aquatic environments [29,30,31,32,33,34,35,36,37,38,39,40,41], including reduction in abundance of susceptible members of microbial communities.

Chemotherapeutants and antimicrobial compounds used in prophylactic treatments have been shown to be effective at reducing or preventing mortalities caused by pathogens [41]. However, some compounds are indiscriminate in their effects, and may also eradicate symbiotic and commensal gut microbial communities [42]. Downstream effects of antibiotic or chemical treatments on microbiomes are likely to have important consequences to fish hosts, and these effects are currently under-studied.

Few studies have documented changes in a fish-associated gut microbial community in response to chemical or antibiotic exposure to externally (topically) applied chemotherapeutants. The effect of ingested antimicrobial compounds on the gut microbiome was widely reported over a considerable period of time in several important aquaculture species, including rainbow trout (*Onchorynkuss mykiss*), using culture methods [43] or molecular-based methods [44]; hybrid tilapia (*Oreochromis niloticus* × *O. aureus*) [45]; and gibel carp (*Carassius auratus gibelito*) [46] (see review in [47]). Collectively, these studies reported that the taxonomic composition and diversity of gut microbial communities were impacted by antimicrobial treatments. These studies, however, focused mainly on describing gut microbiomes in fish at the juvenile stage. Fish at earlier life stages are more prone to pathogen infection [48], and thus may be more frequently exposed to antimicrobial compounds and chemotherapeutants. To evaluate the suitability of prophylactic treatments on fish larvae without compromising fish normal function, more studies are warranted pertaining to the influence of chemotherapeutants utilized in fish culture on gut microbiota.

In this study we characterized microbial community composition and diversity of the larval lake sturgeon gut using 16S rRNA-based next generation sequencing. Lake sturgeon are a species of conservation concern throughout most of their historic range. Where restoration goals to enhance lake sturgeon populations can be met by stocking, streamside rearing facilities (SRFs) are widely used [49]. SRFs utilize a natal water source and are believed to improve the probability of imprinting, compared to traditional hatcheries, which use non-natal well-water for rearing [49,50]. However, the use of SRFs pose challenges, which include increased exposure to temperature fluctuations and spatially and temporally variable surface water (e.g., stream) and hatchery microbial communities [51], including fish pathogens, during early development when mortality is high.

The objective of this study was to quantify and compare gut community diversity and taxonomic composition of larval lake sturgeon raised in an SRF as a function of different chemotherapeutant prophylactics and founding origin. Samples originated from individuals hatched from different egg sources (hatchery vs. wild stream) that were used to quantify the effects of four chemotherapeutants applied prophylactically. We hypothesized that colonization of the gastrointestinal tract would occur during early life stages [52], and that microbial communities associated with different egg incubation environments (hatchery vs. stream) would be reflected in different egg surface community composition and serve as innocula for the gut prior to initiation of chemotherapeutant treatments [5]. We further hypothesized that lake sturgeon larvae treated topically with chemotherapeutants would exhibit decreased GI tract microbial taxonomic diversity and different community composition relative to individuals from a control (no chemotherapeutant) treatment. Detailed effects of microbial founding source and chemotherapeutant treatment will provide insight into the consequences of these effects on host microbe compositional resiliency.

## 2. Methods

### 2.1. Study Site

Use of SRFs and natal water sources, such as the Black River Streamside Rearing Facility (BR-SRF), have been widely advocated in the Great Lakes basin as the preferred method for culturing lake sturgeon [49]. This study was conducted from 26 June to 30 July 2013 at the BR-SRF that is supplied with ambient river water (~680 L/min) from the Kleber Reservoir, located near primary spawning areas for lake sturgeon in the upper Black River in Cheboygan County, Michigan. The mean water temperature recorded during this study was 22.7 °C (min-max 19.9–26.3 °C).

### 2.2. Study Fish

Fish from different egg sources (hereafter called ‘origins’) were employed in this study. The first interaction of bacterial communities and fish progeny occur during early ontogenies even prior to larval hatch at the egg developmental stages [5]. Our previous data indicated that microbial colonization of egg surfaces and the egg microbial succession process is influenced by the community in surrounding water [51,53]. In the context of this study, eggs fertilized and incubated in the hatchery using water pumped from upstream was expected to differ chemically and in terms of biological (e.g., microbial) communities from eggs naturally fertilized and deposited on stream substrate in the natural spawning areas, owing to differences in substrate, groundwater, and surface water influences.

#### 2.2.1. Hatchery-Produced Gamete Collection, Fertilization and Incubation

The purpose of using hatchery-produced larvae was to quantify and compare the effects of different chemotherapeutant prophylactics on gut microbial community diversity and taxonomic composition of a progeny source produced using direct gamete takes, which is commonly utilized in finfish aquaculture including for lake sturgeon [54]. Gametes were collected from two male and two female lake sturgeon spawning in the upper Black River (designated as hatchery family A and B or HA and HB, respectively). Gametes were retained in coolers in the field with an ice pack and transported in plastic bags in river water to the BR-SRF for fertilization to maintain ambient river water temperature. Fertilization took place within four hours of collection. Egg de-adhesion procedures began by applying a Fuller’s Earth solution (Sigma Aldrich, St. Louis, MO, USA) and gently mixing for 50 min using 50 micron-filtered river water. Subsequently, Fuller’s Earth was rinsed from the eggs in 50 micron-filtered river water and at 15 min a 50 ppm iodiphor disinfection treatment was administered. Following a 10 min rinse in 50 micron-filtered river water to remove residual iodiphor using ambient river water, eggs were transferred to Aquatic Eco-Systems (Pentair, Inc., Delevan, WI, USA) J32 Mini Egg-hatching jars for incubation. Beginning two days post-fertilization, eggs were treated daily using a 500 ppm, 15 min bath treatment of hydrogen peroxide until 24 h prior to hatch. After hatch and during the free-embryo period (~7–10 days), lake sturgeon seek refuge in available substrate [55]. Therefore, free-embryos were raised in 10 L polycarbonate tanks (Aquatic Habitats, Inc., Speonk, NY, USA) with a single layer of 2.54 cm^3^ sinking Bio-Balls (Pentair, Inc., Delevan, WI, USA; #CBB1-S) covering the tank bottom. Free-embryo lake sturgeon were raised until endogenous yolk resources were absorbed and fish began a ‘swim-up’ drift behavior (approx. 7–10 days post-hatch). At the onset of exogenous feeding the Bio-Balls were removed and live brine shrimp were provided at 28% body weight three times daily [56].

#### 2.2.2. Field Collection and Incubation of Wild Harvested Eggs and Larval Production

The purpose of using wild, naturally produced larvae for this study was to quantify the effects of different chemotherapeutant prophylactic treatments on gut microbial community diversity and taxonomic composition of an additional progeny source utilized in sturgeon aquaculture [54]. Naturally produced, fertilized eggs were collected from stream substrate in the Upper Black River at two spawning site locations approximately three days post-fertilization. Eggs were transported to the BR-SRF in river water and incubated, separated by capture location (wild site B, and site C and designated as WB and WC), in Aquatic Eco-Systems (Pentair) J32 Mini Egg-hatching jars. Eggs were treated daily using a 500 ppm, 15 min bath treatment of hydrogen peroxide until 24 h prior to hatch. After hatch and during the free-embryo period, lake sturgeon were reared in the BR-SRF under conditions described above as originally developed for Michigan State University Animal Use and Care standard operating procedures and subsequently published [56].

### 2.3. Experimental Treatments

Details concerning the experimental design including descriptions of facilities and equipment used to conduct the experiment and background to the major independent variables (hatchery or wild sample source and chemotherapeutant treatments) are provided in Figure 1 and below. At twelve days after initiation of exogenous feeding, we transferred 400 fish from each hatchery origin family (HA and HB) and each stream spawning origin group (WB and WC) into four 1.2 m diameter tanks, which were divided into eight partitions (50 fish per partition). Filtered (50 micron) river water was used in all tanks to remove large particulates and aquatic invertebrates and fish. Each partition was randomly assigned to one of four weekly chemotherapeutant treatment types, each with two replicates (Figure 1). The study began at fourteen days post-exogenous feeding after a two-day tank acclimation period, and continued for thirty-five days to quantify and compare the effects of different prophylactic chemotherapeutants on gut microbial community diversity and taxonomic composition. Chemotherapeutants administered in this study included those commonly utilized in traditional hatcheries and SRFs. Weekly prophylactic treatments in this study included: (1) 60 min, 15 parts per million (ppm) CT bath; (2) 15 min, 60 ppm H_2_O_2_; (3) 3 parts per thousand (ppt) NaCl_2_ bath for 15 min followed 24 h later by a 15 min, 60 ppm H_2_O_2_ bath; and (4) a control (no chemical treatment). Fish were fed three times daily as described above, except on treatment days when feeding was delayed until all treatments had been performed. Each week, all fish from each treatment type (including no treatment controls) were transferred using a small aquarium dip net that was unique to each tank and section, to 10 L polycarbonate tanks equipped with one aerator in each tank. Fish were administered respective treatments, briefly rinsed in 50 micron-filtered river water, and placed back into their rearing tank. All treatments were administered on the same day, once per week except treatment 3, which included an additional treatment the following day with H_2_O_2_. Controls were handled in the same manner as all other treatment groups; however, similar to treatment 1, were held for 60 min in their ‘treatment’ tank before being rinsed and returned to their rearing tanks. Mortalities were removed from the tanks each day and recorded to quantify survival at the end of the study. The duration of this experiment lasted thirty-five days (forty-nine days post-exogenous feeding) to encompass the period of high mortality documented in SRFs.

Sampling for microbiota analysis took place following the end of the five-week treatment period the day following the last chemotherapeutant exposure. From each partition (*n* = 2), four fish were randomly collected (*n* = 4), and were euthanized with an overdose of MS-222 (Sigma-Aldrich, St. Louis, MO, USA) according to Michigan State University IACUC-approved animal use and care protocols. All fish (*n* = 128) were preserved in 80% ethanol and transported to MSU until dissections were performed within one month of collection.

### 2.4. Fish Dissection, DNA Isolation, PCR Validation

The distal gut (spiral valve) of each lake sturgeon larvae was recovered from fish following aseptic techniques. The distal gut was defined as the section that includes the end of the intestine through the distal end of the spiral valve. The spiral valve serves as the primary region of digestion and absorption, and thus may provide an area of abundant nutrients where a microbial community can flourish [57,58]. Exterior surfaces were swabbed with 100% ethanol before dissections of the whole digestive tract using sterile instruments. Dissections were performed with slight modification, as previously described by [59]. The intact alimentary tracts were cut from the body cavity, and the excised gut was immediately transferred into filtered-sterilized 80% ethanol solution for DNA isolation. All dissected samples were stored in −20 °C for <1 wk until DNA extractions were performed following the dissection.

Each gut sample was first centrifuged at 12,000× *g* rpm for 15 min at 4 °C to pellet bacteria that may have leached from the sample before DNA was extracted. The combined gut and pelleted bacteria were extracted using The MoBio PowerSoil^®^ DNA Isolation Kit (Carlsbad, CA, USA) including a bead-beating step, following protocols for low-biomass samples, as suggested by the manufacturer. The integrity of each DNA sample was assessed based on amplification of 1.4k bp of the 16S rRNA gene (amplicon based on 27F and 1389R primers) followed by gel agarose electrophoresis (1% agarose in TAE buffer). DNA concentrations were quantified by absorbance at 260 nm in a Microplate spectrophotometer (BioTek^®^, Winooski, VT, USA).

### 2.5. 16S rRNA Amplicon Sequencing and Sequence Pipeline Analyses

Gut microbiota from lake sturgeon larvae were surveyed using high-throughput sequencing of the V4 region of the 16S rRNA gene. In total, 152 DNA samples (over four treatments, sampled at three time periods, including four positive controls, water samples, and technical replicates; see Figure 1 and description in Section 2.3) that had been validated to contain sufficient bacterial DNA (as shown by the presence of amplicon bands in electrophoresis) were submitted for sequencing at Michigan State University Research Technology Support Facility, (RTSF—(https://rtsf.natsci.msu.edu/genomics/ (East Lansing, MI, USA, accessed on 20 August 2014)). All sequencing procedures, including the construction of the Illumina sequencing library, emulsion PCR, and MiSeq paired-end sequencing v2 platforms of the V4 region (~250 bp; primer 515F and 806R) followed standard Illumina (San Diego, CA, USA) protocols. Michigan State Genomics RTSF provided standard Illumina quality control, including base calling by Illumina Real Time Analysis v1.18.61, demultiplexing, adaptor and barcode removal, and RTA conversion to FastQ format by Illumina Bcl2Fastq v1.8.4. Raw sequence reads were deposited to the NCBI Sequence Reads Archive (SRA) under BioProject accession number PRJNA820564 (accessed on 28 March 2022).

Details of the microbial sequence data analyses pipeline and computing workflow were made following the suggested settings of mothur’s operation protocol (https://www.mothur.org/wiki/MiSeq_SOP, accessed on 28 March 2022). Briefly, paired-end sequence merging, quality filtering, “denoising”, chimera checking, and pre-cluster steps were conducted using an open-source workflow based on methods implemented by program mothur v.1.42 [60]. Sequence pipeline analyses were performed in mothur v.1.42 to accomplish reference-based OTU clustering (method = opticluster). Taxonomic assignment was performed by first aligning sequences data using the SILVA 132 bacterial reference database followed by clustering sequences defined with 97% identity and later classified using Ribosomal Database Project (RDP) 16 (V5.4) training set. Given the length of retained sequences, Operational Taxonomic Unit (OTU) criteria representing sequences that are not more than 3% different from each other, and our desire to compare data presented here to previous gut microbiome research (e.g., [52]), we chose to define taxonomic variation based on OTUs rather than Amplicon Sequence Variant (ASVs). Any sequence singletons that were detected were removed prior to downstream analyses. Rarefaction analyses were performed to evaluate the coverage for each sample based on the selected sequence depth. To minimize effects of under-sampling while maintaining as broad a dataset as possible, the final OTU table (Appendix A) was rarefied to a depth of 10,000 sequences per sample. Nine DNA samples with low sequence depth were discarded prior to downstream analyses. The community matrix describing sequence counts for all OTUs for all treatments associated with this study can also be found on GitHub at https://github.com/ScribnerLab/Chemotherapeutants.git (doi.org/10.5281/zenodo.6418537, accessed on 21 April 2022).

### 2.6. Statistical Analysis of Bacterial Community Profiles and Ecological Statistical Analyses

#### 2.6.1. Alpha Diversity

Measures of microbial community diversity including inverse Simpson (1/D) diversity indices and OTU richness for each sample from larvae from each chemotherapeutant treatment and origin (wild and hatchery egg sources) were calculated from community matrices derived from program mothur based on sequence data. All statistical analyses were carried out in the R program (v3.0.2).

Diversity indices (inverse Simpson and OTU richness) were first evaluated using a Kruskal–Wallis non-parametric test for control groups (no chemotherapeutant added) to determine whether there were statistical differences that existed in unperturbed microbial community alpha [α] diversity measures in the lake sturgeon larval GI tracts as a function of egg origin (wild vs. hatchery). The test was performed instead of parametric tests that assume a normal distribution. Next, the effects of chemotherapeutants and egg origin on measures of microbial gut community diversity were estimated based on a generalized linear model (GLM) using suitable probability distributions (inverse Simpson = Gamma distribution; Richness = Quasipoisson distribution) in R program (v3.0.2) using glm(). The GLM method has been shown to have high efficiency when estimating parameters, yielding interpretable estimates that also avoid transformation bias [53,54]. *p*-values < 0.05 indicated significance of the effect of variable on alpha diversity measures. Relative abundance estimates of bacterial phyla in all fish gut and water-associated microbial community samples at the end of the fifth and final treatment was determined using packages dplyr and reshape2 in program R (v3.0.2).

#### 2.6.2. Beta Diversity

We included several packages implemented in program R to estimate (beta [β]) diversity measures quantifying bacterial community compositional differences between samples and ecological statistics at the bacterial OTU level. Briefly, vegan [61] was used to produce a Bray–Curtis (BC) [62] dissimilarity matrix, and to perform non-metric dimensional scaling (NMDS) ordination as a means of characterizing differences in microbial community composition among samples. We used the nmds function to perform non-metric dimensional scaling (NMDS) ordination to visualize community compositional differences based on sample BC dissimilarity. The ggplot and ggplots2 packages [63] were used to create ordination plots to visually compare sample gut community composition as a function of different treatments, between sampling origins, and water samples.

Next, we performed multivariate hypothesis testing to quantify differences in community composition among samples originating from different groups based on locations of egg origins and exposed to different chemotherapeutant treatments using the adonis function [61] in program R (v3.0.2). Two different fish families (hatchery origin) and two river spawning locations (wild origin) were treated as replicates. Analyses focused on the effects of chemotherapeutant treatments and origin. Permutational multivariate analyses of variance (PERMANOVA) was conducted on BC dissimilarity matrices of fish associated microbial community composition [64,65]. Under the null hypotheses, the centroids of the groups (fish from either hatchery and wild groups that were exposed to different chemotherapeutant treatments) were expected to be equivalent for all groups under random allocation (i.e., based on permutation) of individual sample units to the groups.

Analyses investigated whether host origin and/or chemotherapeutant treatment had a significant effect on microbial community structure. NMDS and PERMANOVA were performed on fish gut communities within each origin group to determine whether chemical treatments had effects on fish gut microbiota. Under the null hypothesis, chemotherapeutant treatments were not expected to significantly affect fish gut community taxonomic composition within an origin group, in part because eggs from both hatchery and wild origins were exposed to peroxide during incubation that was believed to reduce and taxonomically homogenize samples for all treatments and both origins. PERMANOVA analyses that indicated significant treatment effects were then analyzed using post hoc tests using betadisper and permutest functions followed by a Tukey test to determine which treatment(s) differ significantly in larval lake sturgeon gut bacterial taxonomic composition.

#### 2.6.3. Differential Abundance of OTUs and Biomarker Identification across Treatments

To determine the operational taxonomic units (OTUs) that most likely explained differences in microbial larval lake sturgeon gut community composition between fish from different origins and among different chemotherapeutant treatment groups, we next employed linear discriminant analysis (LDA) effect size (LEfSe) methods [66]. In general, the LEfSe algorithm identifies genomic features (i.e., bacterial OTUs) that were differentially abundant in different experimental groups (origin groups and treatments), then ranks them based on that abundance differential. The larger the difference in relative abundance between groups, the higher the importance of that OTU. 

The algorithm first identified features (OTUs) that were statistically different among origin groups based on the nonparametric factorial Kruskal–Wallis (KW) rank sum test. Additional tests assessed the consistency of differences using unpaired Wilcoxon rank sum tests. In the final step, LEfSe used LDA to rank each differentially abundant taxon in order of the difference in abundance based on an LDA Score (log-scale). Results represent a scale indicating “importance” of an OTU in origin group differences in microbiota composition [66].

To run LEfSe, a tabular file was generated from a shared file that contained no singletons in the program mothur v.1.39.5. The tabular file consisted of taxonomic relative abundance in gut community samples from the four different origin samples that were all exposed to four chemotherapeutant treatments. This tabular file was transferred using an online bioinformatics toolkit developed by the Huttenhower lab to perform LEfSe analyses (https://huttenhower.sph.harvard.edu/galaxy/, accessed 20 November 2014).

## 3. Results

### 3.1. Diversity of Gut Microbial Community Composition 

A total of 144 samples were retained after quality filtering was performed in the sequence pipeline analyses. Comparisons of lake sturgeon larvae gut microbial community composition at the level of phyla indicated that three major phyla dominated more than 65% of total community abundance across all fish samples (*Firmicutes* 16%*, Proteobacteria* 36.5%*,* and *Actinobacteria* 15.1%). Phyla detected in the remainder of the gut community included *Acidobacteria*, *Bacteroidetes*, and *Verrucomicrobia* that collectively comprised 30% of gut communities. 

The relative abundance of the most dominant phylum, *Firmicutes*, was fairly consistent across treatments for fish samples from all hatchery and wild origin groups and wild groups (HA, HD and WB, WC, respectively). One exception was WB larvae exposed to salt (mean 58%) and WC fish exposed to peroxide (mean 50%) that were relatively low compared to other treatments (Figure 2a). When comparing the abundance of *Firmicutes* across all groups, fish from hatchery family D (HD) had a lower percentage of *Firmicutes* (mean range from 51–66%). *Proteobacteria* relative abundance was likewise relatively uniform across treatments (13–28% of total abundance) with the exception of WB fish that were treated with chloramine-T, CT (6%). *Actinobacteria* were present at 1% in fish that were not exposed to any chemotherapeutant (control) and only in fish from HA and WC origin groups. At the genus level, Firmicutes were represented by two genera, *Clostridium_sensu_stricto* & unclassified genera from family *Clostridiaceae.* We found that *Clostridium_sensu_stricto* were the most dominant genus (mean range: 30–51% of the total community) for all fish of hatchery origin (except for HA fish exposed to peroxide), whereas all fish of wild origin had unclassified taxa from *Clostridiaceae* family (mean range: 29–62%) as the most abundant genus across any treatment (Figure 2b). Genera from phylum *Proteobacteria* including several unclassified taxa from *Betaproteobacteria*, unclassified taxa from *Enterobacteriaceae*, unclassified taxa from *Rhodobacteriaceae*, and *Deefgea* all were present at lower percentages of abundance with more amounts of variation across fish groups and treatments (Figure 2b). The only genus in the phylum *Actinobacteria* that was detected among dominant taxa was the genus *Zhihengliuella*, present in HA control fish (mean 2.2%) and WC control fish (mean 1.4%).

Figure 3a,b revealed results of GLM tests comparing inverse Simpson indices and a number of observed taxa among chemotherapeutant treatments and origin groups. As opposed to our initial hypothesis, fish in the control treatment (CT) had less diverse gut communities (both inverse Simpson and richness) with the exception of fish in family HD. Fish exposed to salt treatment were characterized by higher inverse Simpson and greater taxa richness than communities from samples exposed to other chemotherapeutant treatments in wild family, WB. For individuals from wild family WC, we found that fish exposed to peroxide had a greater number of taxa relative to fish from wild family WB from the control group (Figure 3b). Our analyses did not quantify family effects since families (hatchery origin) and stream locations (wild origin) served as replicates for each origin. We observed large heterogeneity among samples collected following different treatments and between egg origins (Figure 3a,b). For example, differences between communities sampled from individuals from the control and salt treatment groups associated with hatchery family HD and wild family WB were observed for Simpson’s inverse diversity and were higher for samples in the peroxide and salt treatments. To summarize, Kruskal–Wallis tests for taxa richness and inverse Simpson among fish from control groups indicated that no significant difference existed between groupings based on egg origins (*p* > 0.05). Statistical analyses based on the generalized linear model (GLM) indicated that gut communities of individuals exposed to certain treatments (salt and peroxide) had significantly different levels of taxa richness, but not on the inverse Simpson indices (Appendix A).

### 3.2. Differences in Gut Microbial Community Composition between Fish Group Origin and among Chemotherapeutant Treatments

Non-metric dimensional scaling (NMDS) ordination of BC dissimilarities in microbial taxonomic composition of gut communities was performed to visualize community compositional relationships among larval gut samples associated with fish from different egg origin and exposed to different chemotherapeutant treatments. Four NMDS plots were generated, including Figure 4a: all fish gut microbiota; Figure 4b: gut microbiota for fish in control treatment groups only; Figure 4c: gut microbiota community relationships among chemotherapeutant treatments for fish originating from a hatchery (two families, HA and HD); and Figure 4d: gut microbiota for fish among chemotherapeutant treatments originated from the stream substrate (wild groups from two spawning locations; WB and WC). All ordination plots were characterized by stress values ~0.2 indicating that data were well represented in 2D NMDS plots. Community membership across samples of similar origin (either from the wild, or from the hatchery production) were clustered together regardless of treatment groups as denoted by the ordination pattern suggesting influence of egg origins on fish gut microbiome (Figure 4a). Baseline community membership in fish without any chemotherapeutic treatment (control group) was visualized in Figure 4b, revealing that fish from eggs collected from the wild (WB and WC) exhibited considerably higher inter-sample variation in community composition relative to the variation among fish originating from hatchery crosses (HA and HD) across all chemotherapeutant treatments. 

To quantitatively test for gut community compositional differences among chemotherapeutant treatment and origins, PERMANOVA was performed. Comparisons of microbial OTU beta diversity across samples from the controlled groups indicated that gut microbial communities from control groups were not significantly influenced by the egg origin (Table 1). Subsequently, the effects of chemotherapeutant treatments on fish gut microbiomes were investigated across all samples taking into consideration both the effects of chemotherapeutant treatment and where the fish originated from (hatchery vs. wild). No influence of chemotherapeutant treatment was detected, but the effects of egg origin were significant (Table 2). 

Additional analyses of chemotherapeutant effects on fish gut microbiome composition were investigated separately based on fish origin (hatchery vs. wild). Chemotherapeutant treatments had significant effects on larval gut microbiomes between individuals from different hatchery families (HA, HD) as indicated by PERMANOVA test results (Table 3; *p* = 0.012). However, the effect of chemotherapeutant treatment was not evident between fish from eggs collected in different regions of the stream (WB, WC), although a significant interaction was observed between origin group and treatment (Table 4).

To better understand the effects of different chemotherapeutants in gut communities from hatchery fish, post hoc tests, betadisper and permutest, were conducted followed by Tukey’s test. The adjusted *p*-value from Tukey’s test indicated that none of the communities associated with different treatments differed statistically, although betadisper revealed that the distance of each point to the centroid for salt and peroxide differed.

### 3.3. Identification of Bacterial Taxa Influenced by Chemotherapeutant Treatments

Given findings of effects of origin and treatment on microbial community beta diversity, we used LEfSe to identify which taxonomic groups showed the largest differences in relative abundance when fish from the same origin were exposed to treatments (Figure 5). We first compared microbial communities from fish from the control groups from hatchery and wild origins at the genus level (all vs. all). We found taxa associated with phylum Actinobacteria, including genus *Methylocystis* from phylum *Firmicutes*, and genus *Salinicoccus* from phylum *Proteobacteria* differed in abundance (LDA score higher than 2.0, *p* < 0.05, see Figure 5a) for the comparison between fish communities in the control group between both egg origins. These three genera were present in higher abundance in hatchery fish samples as opposed to wild fish.

We likewise compared communities of fish from the control treatment within each origin (wild and hatchery, respectively) to other chemotherapeutant treatment groups (one vs. all). LEfSe analyses performed with fish from the wild group detected two differentially abundant taxa associated with genus *Clostridium_XVIII* (Phylum *Firmicutes*) and *Methylocystis* (Phylum *Proteobacteria*). Both genera were present in high abundance in the guts of fish exposed to the peroxide treatment, and for genus *Clostridium_XVIII*. The taxa were also abundant in the guts of fish from the salt treatment (LDA score higher than 2.0, *p* < 0.05, see Figure 5b). For hatchery origin fish, LEfSe analyses on fish that were treated with peroxide revealed the presence of genera *Kocuria* and *Nocardia* (both from phylum *Actinobacteria*) in high abundance, while fish exposed to the salt treatment had *Peptoniphlus* and *Luteimonas* that were in higher abundance compared to individuals from other treatments (see Figure 5c).

## 4. Discussion

### 4.1. General Findings and Relevance to Aquaculture

Understanding interactions between microbes and the host surfaces they colonize is important to aquatic animal health and aquacultural production [5,35]. Potentially harmful changes can occur to beneficial gut microbes from the over-utilization of chemotherapeutants [34,67], which can result in ecological drift or selective community alteration that can favor increases in the abundance of undesirable taxa [68,69,70]. Additional adverse effects related to antibiotic use include pathogen resistance, suppression of the immune system, increased rates of allergies, autoimmunity, and other immune-inflammatory conditions [34]. Microbial community changes anatomically and ontogenetically in response to spatial and temporal environmental variation, and changes related to perturbations have also been described in fish taxa [5], but are less well investigated. In well studied humans, microbiomes within individual hosts usually vary in composition across anatomical sites, and microbial taxonomic composition can vary over time in response to factors such as diet, physical activities, and medication intake [32,71,72].

In this study, we found little evidence for the influence of commonly used chemotherapeutant treatments applied topically in water baths to larval lake sturgeon prophylactically on gut microbiome composition. Data did indicate greater influence of microbial founder effects (hatchery vs. wild stream origin), which may be explained by exposure to environmental sources during earlier life stages or influences of genetic effects [73,74]. Results could also indicate genetic or maternal effects reflecting different family membership of fish from different origin groups. We provide an interpretation of origin and chemotherapeutant treatment results and discuss implications for aquatic animal health in aquaculture generally.

### 4.2. Effects of Chemotherapeutant Treatments on Larval Gut Microbial Communities

All chemotherapeutants used in our study are commonly used for the treatment of external pathogens rather than orally administered to fish. In fish aquaculture, prophylactic treatments are widely used to control pathogenic bacteria disease outbreaks that commonly occur in hatcheries during vulnerable early life stages. Chloramine-T and peroxide are widely used to control and eliminate infection associated with flavobacteriosis [8]. Overall, our results indicate that chemotherapeutant treatments during larval stages did not result in large changes in the composition of the intestinal microbiota, at least during the short observation and experimental period (five weekly exposures). Although GLM suggested that taxa richness may be significantly influenced by certain treatments applied, such as salt and peroxide, the same treatments did not have a significant effect on the inverse Simpson indices. PERMANOVA and least square means tests revealed that the chemotherapeutant treatments employed in our study had only a minor effect on intestinal gut microbiome in lake sturgeon larvae; although effects varied among fish with different backgrounds associated with families and their sampling origin.

There are several potential explanations for the comparatively small effects of chemotherapeutant treatments on larval gut microbial communities. One explanation is that the externally administered treatment did not enter the digestive tract in significant enough concentrations or duration to alter the gut community. When larval fish are provided chemotherapeutants prior to feeding, rather than during feeding, microbial compositional stasis suggested that the chemicals may not enter the gastrointestinal tract. Alternatively, the effect of the treatment may not have been evident due to the short treatment duration (15–60 min bath immersion) and weekly periodicity of chemotherapeutant treatments. Exposure to chemotherapeutants, consistent with our methodology, may not have been of sufficient concentration to result in quantifiable changes in gut community composition. In addition, fish were returned into their tank partition after treatments, and that may have allowed rapid recolonization of gut microbiota from the surrounding water. Further, chemotherapeutant treatments were administered at seven-day intervals, potentially allowing community recovery. The microbial communities may have exhibited resiliency to chemotherapeutant treatment; returning to a similar compositional state during the several day period between the timing of treatment and sampling for gut interrogation. Further studies are warranted to quantify the amount of any compound entering the gut during the treatment period to ascertain causal relationships.

In the LEfSe analyses, three out of thousands of microbial taxa appeared to be tied to differences between untreated fish in a hatchery and the wild. After fish were exposed to chemical treatments, different taxa were reported to be differentially abundant. Those taxa, however, are not among the dominant taxa. It is unclear how treatment differentially affected the relative abundance of these taxa. Results could indicate that the gut microbiota were either resistant or exhibited resilience in community composition, where treatment-based changes were short-lived and communities rapidly returned to their original state [30]. The communities could also have had different compositional taxonomy, yet were still able to maintain function (functional redundancy). Navarrete and colleagues [75] focused on determining the effects of a dietary inclusion of *Thymus vulgaris* essential oil (TVEO) on microbiota composition, compared with a control diet without TVEO over a 5 week period. Their study indicated high similarities between gut microbiota in treated and non-treated fish, and TVEO induced negligible changes in gut microbiota profiles. Essential oils include volatile liquid fractions produced by plants that contain the substances usually responsible for defenses against pathogens and pests due to their antibacterial, antiviral, antifungal, and insecticidal activities [76]. We conclude, based on LEfSe results (Figure 5b,c), that gut microbiota composition in lake sturgeon was persistent and stable throughout the trial, producing relatively consistent molecular profiles.

We detected three major phyla *Firmicutes*, *Proteobacteria*, and *Actinobacteria*, that dominated the lake sturgeon larval gut community across all samples (Figure 2). The most predominant taxa that were detected from phyla *Proteobacteria* (such as *Enterobacteriaceae, Rhodobacteriaceae*) are Gram-negative bacteria. Many studies have shown that Gram-negative bacteria are resistant to commercially available antibiotics partly due to their thick cell wall structure compared to Gram-positive bacteria [77,78]. *Enterobacteriaceae* include a group of bacteria known as extended-spectrum beta-lactamase (ESBL) *Enterobacteriaceae* that can confer resistance to antibiotics via production of the β-lactamase enzyme, which can inactivate certain β-lactam antibiotics [79].

Another major phylum, *Firmicutes* that were detected in fish guts across all families and treatments was primarily represented by Unclassified *Clostridiaceae1* and taxa *Clostridium sensu stricto*. Although *Clostridia* are Gram-positive, these bacteria have been identified as part of commensal gut microbiota that plays major roles in the maintenance of the gut homeostasis. Several features associated with *Clostridium* spp. could explain why this taxon can thrive in the gut and can likewise be resistant to prophylactic treatments administered in our study. In humans, *Clostridium* spp. are involved in defenses inside the intestinal microecosystem along with gut-associated lymphoid tissue (GALT), and confer resistance against pathogen infections. This taxon is thought to have immunological tolerance [80]. In addition, cultured *Clostridium* spp. exhibit the ability to form endospores, which offers this bacteria ecological advantages for survival under adverse conditions [80,81].

Comprehensive studies on adverse effects of antibiotic use to the gut microbiomes were reported in other fish species [5,37,39] and in humans [33]. Exposure to antibiotics can have profound effects on resident microbial communities inside human guts [34,72]. Several studies reported changes in density or gut microbiome composition, for instance in human infants who receive antibiotics [82]. Dethlefsen et al. [6,7] documented the pervasive effects of an orally administered antibiotic to adult gut microbiomes, associated with decreases in taxa richness and evenness and can lead to community changes in composition and function [33].

Relatively few studies have been conducted addressing the effects of chemotherapeutants administered topically in water baths on fish gut microbial communities as conducted in this study. Most studies have been conducted on salmonids or tilapia [43,45,75] and gibel carp (*Carassius auratus gibelito*) [46], and have focused on the effects of antibiotics administered orally to address infection levels of known pathogenic bacteria. Navarrete et al. [44] reported that gut microbiomes of salmonids exposed to the antibiotic oxytetracycline (OTC) that was orally administered were characterized by lower taxonomic diversity and were primarily composed of *Aeromonas.* The results were consistent with findings from another study conducted to evaluate the effects of orally administered antibiotics to gibel carp [46]. Importantly, the results from studies using orally administered antibiotics different from our data.

### 4.3. Sources of Heterogeneity Associated with Microbial Community Origin

A prerequisite for developing a strategy for microbial pathogen control is a knowledge of resident aquatic microflora associated with fish larvae, and how interactions between larvae and microflora occur. De Schryver & Vadstein [83] suggested that the primary means by which pathogens could be controlled is the water surrounding animals.

Fish produced from wild eggs show greater community diversity compared to artificially produced fish in the hatchery (Figure 4b). Thus, the initial inoculation location on the egg chorion surface likely determined their community during later life stages, as community successional changes occurred [5]. Alternatively, gut microbial communities in wild fish may have exhibited greater resilience to treatments and maintained their gut compositional similarity. In contrast, hatchery fish originated from eggs that have been artificially produced in hatchery facilities; therefore, they had limited contact with their respective natural habitat like the wild eggs, except their egg surfaces reflect aquatic communities where their parents spawned (in the hatchery). This could also suggests that domestication selection, in terms of hatchery gut community establishment, occurs in fish produced in a hatchery, affecting the community structure of their gut microbiome.

In fishes, microbial binding to host cell surfaces is often mediated through the interactions of bacterial carbohydrate binding proteins (lectins) with host cell surface carbohydrates [84,85]. Stream substrates are extremely variable and likely harbor different microbial communities than are present in stream water used in stream-side or traditional (often ground water) hatchery facilities. Different microbial communities have been characterized from naturally spawned lake sturgeon eggs in the WC and WB areas of the upper Black River previously (Marsh unpubl. data). Larvae hatching from eggs deposited on stream substrates typically remain in close proximity to egg surfaces for long periods when gill surfaces likely acquire and internalize egg surface-bound microbial taxa. If this period is indeed the point at which larvae internalize egg surface-bound microbial taxa, then this occurs prior to the full development of alimental structures [5]. Thus, differences in founding microbial communities between hatchery and wild sources are probable. This source of heterogeneity and subsequent successional changes in community diversity and composition can be important for later life stages of fishes.

Several studies of gut microbiota in fish with different genetic backgrounds have documented that host genotype (broadly defined) may contribute to compositional heterogeneity among individuals in fish gut microbiota, at least to some extent. Abdul Razak et al. [53] studied catfish gut microbiome assembly and quantified changes in gut microbiome development from eggs to stock-out juveniles released into nursery ponds. The study identified host genotype (families), dietary factors, and environmental (rearing pond) effects. Significant differences in alpha diversity were evident at the egg stage, yet the differences diminished as fish matured. The authors found evidence of significant interactions between family and stocking pond environment on larval gut microbiota composition, as was also found in this study.

Another study [86] demonstrated evidence of host effects on the intestinal microbiota of captive and wild whitefish. Whitefish (*Coregonus* spp.) species pairs and their reciprocal hybrids were reared in captivity under a controlled environment. Analyses revealed significant effects of the host genetic background on the taxonomic composition of the transient microbiota. Navarrete et al. [87] assessed the relative effects of a host (genotype) and diet to gut microbiome composition of rainbow trout (*Onchorhynchus mykiss*). Full-sibling fish from four non-related families were fed two diet regimes in comparison to the control group. Results showed that some relative abundance of several bacterial taxa differed among trout families, indicating that the host genotypes may influence gut microbiota composition. In addition, the authors reported that the effect of diet on microbiota composition was dependent on the trout family. Studies on other organisms, such as chickens, also showed that under a common diet and husbandry practices, gut microbiota composition differed between two lines (high weight, HW and low weight, LW) [88]. Findings from Blekhman et al. [89] indicate that human gut microbial variation are driven by host genetic variation involving genes that have been previously associated with microbiome-related complex diseases. They also showed that host genomic regions associated with microbiomes have high levels of genetic differentiation among human populations, suggesting host-genomic adaptation to environment-specific microbiomes. This finding could be possibly true for fish as well where variation in gut microbiome is attributed to genetic background.

## 5. Conclusions

Findings in this study detail observed differences in microbial founding sources (water borne and substrate specific egg microbial incubation environments) and chemotherapeutic treatments to developing microbial communities during early ontogenetic stages. These results provide an insight into the consequences of prophylactic treatments and host-microbe interactions. Our study serves as a baseline providing information on the indirect effects of chemotherapeutant intervention that could either positively or negatively affect the normal gut microbiota. Results of minor effects associated with use of chemotherapeutants prophylactically suggest that topical use at the ontogenetic stage and concentration used may not have negative indirect effects on resident gut microbial communities. Thought should be given to the selection of locations to collect gametes to bring into culture. Future work could profitably focus on identifying microbial taxa that colonize the external surfaces of the fish (gill plate, gills, ventral area between pectoral fins, etc.) to see how external treatments impact the colonization of external microbes. Further studies are also warranted that would compare the effects of treatments when administered following pathogen infection.

## Figures and Tables

**Figure 1 microorganisms-10-01005-f001:**
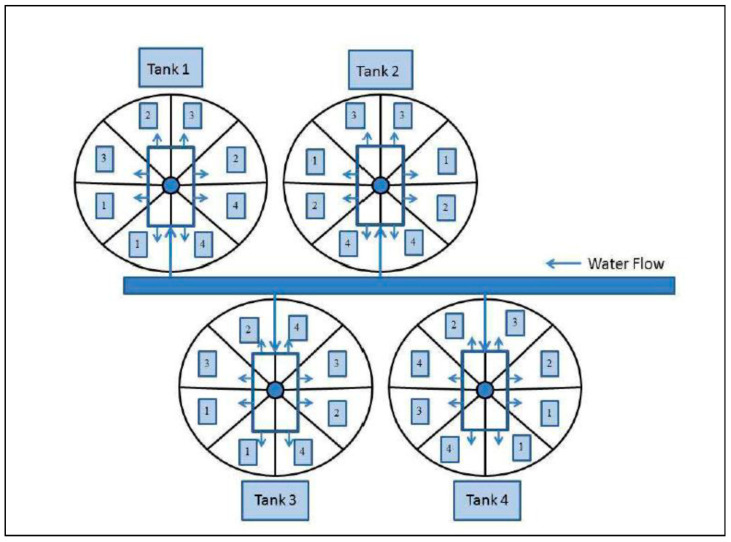
Schematic design of the larval chemotherapeutant study. Each 1.2 m diameter of tank held 400 fish from hatchery and wild naturally produced fish, which were divided into eight equal sized partitions (50 fish per partition). There were four tanks. Each partition was randomly assigned to one of four weekly treatment types, each with two replicates. Chemotherapeutant treatments included: (1) 60 min, 15 ppm CT bath; (2) 15 min, 60 ppm H_2_O_2_; (3) 3 parts per thousand (ppt) NaCl- bath for 15 min followed 24 h later by a 15 min, 60 ppm H_2_O_2_ bath labeled as NaCl/H_2_O_2_; and (4) a control (no chemical treatment) labeled as CTRL. Arrows indicate directions of water flow.

**Figure 2 microorganisms-10-01005-f002:**
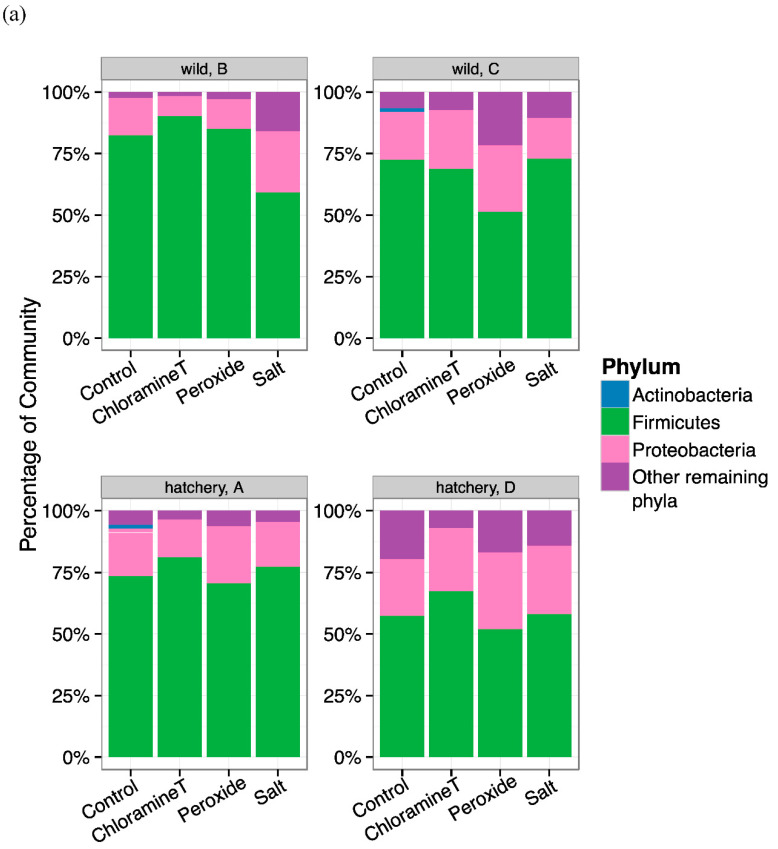
Taxonomic composition of bacterial communities identified from the lake sturgeon larval GI tracts (**a**) at the phyla level and (**b**) at the genera level. (**a**) Relative abundance (percentage) of dominant bacterial phyla found in the gut microbiota of lake sturgeon larvae separated based on sample family/group to display variation in communities across prophylactic treatments. Three predominant phyla were present in gut microbial communities (*Firmicutes*, *Proteobacteria*, *Actinobacteria*). The other phyla were characterized as Others; (**b**) relative abundance (percentage) of dominant bacterial taxa found in fish gut samples, separated by family/group and treatment. Among the most abundant taxa included *Unclassified Betaproteobacteria*, *Unclassified Clostridiaceae_1*, *Clostridium_sensu_stricto*, and *Unclassified Enterobacteriaceae*.

**Figure 3 microorganisms-10-01005-f003:**
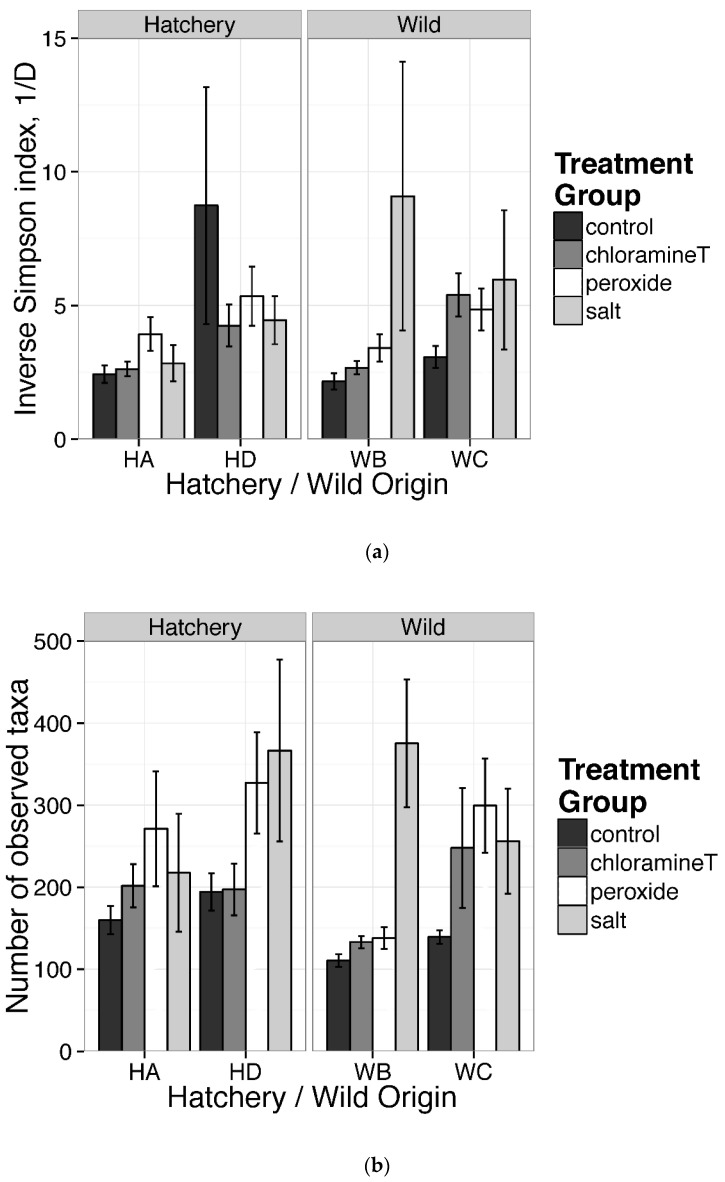
Estimates of alpha diversity: (**a**) inverse Simpson index; (**b**) number of observed taxa (OTU richness) for lake sturgeon gut microbial communities from all samples of treatments and families within location of origins. Each bar indicates mean with S.E. for each treatment from each family within origins.

**Figure 4 microorganisms-10-01005-f004:**
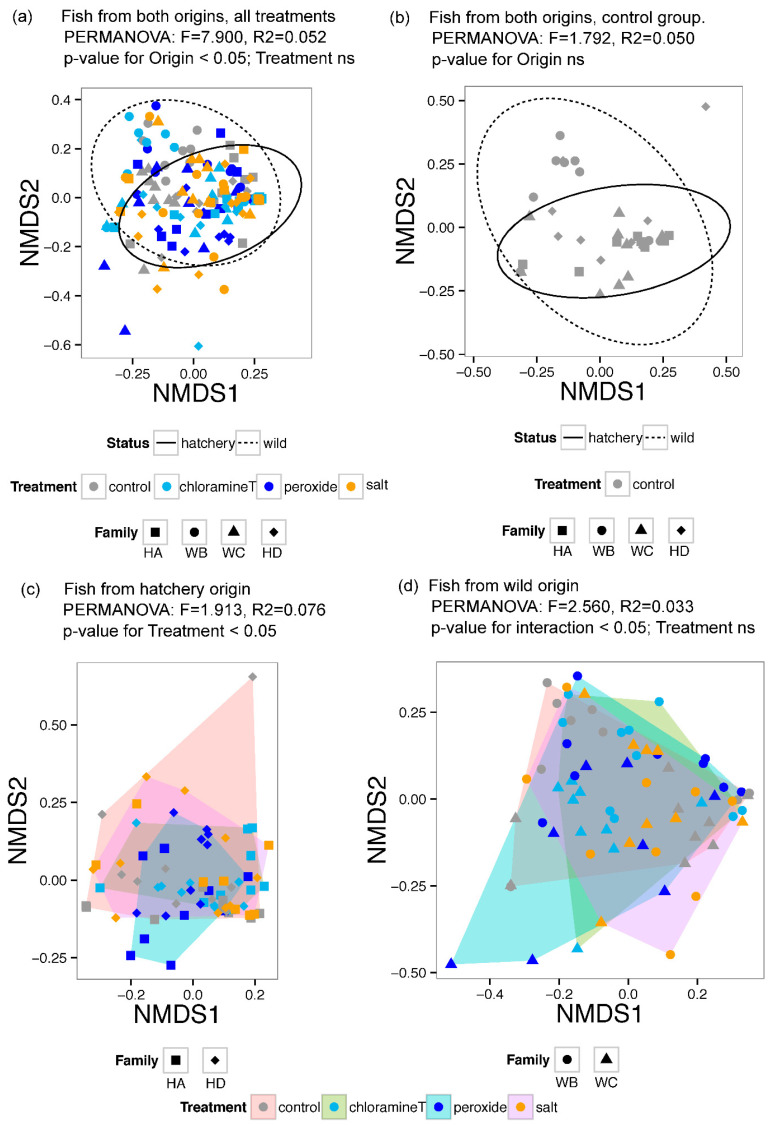
Non-metric dimensional scaling (NMDS) ordination of Bray–Curtis differences in larval lake sturgeon gut microbial taxonomic composition (**a**,**b**) and results of accompanying PERMANOVA analyses (**c**,**d**) characterizing relationships among larval gut samples associated with fish from different groups associated with egg origin and those exposed to different chemotherapeutant treatments.

**Figure 5 microorganisms-10-01005-f005:**
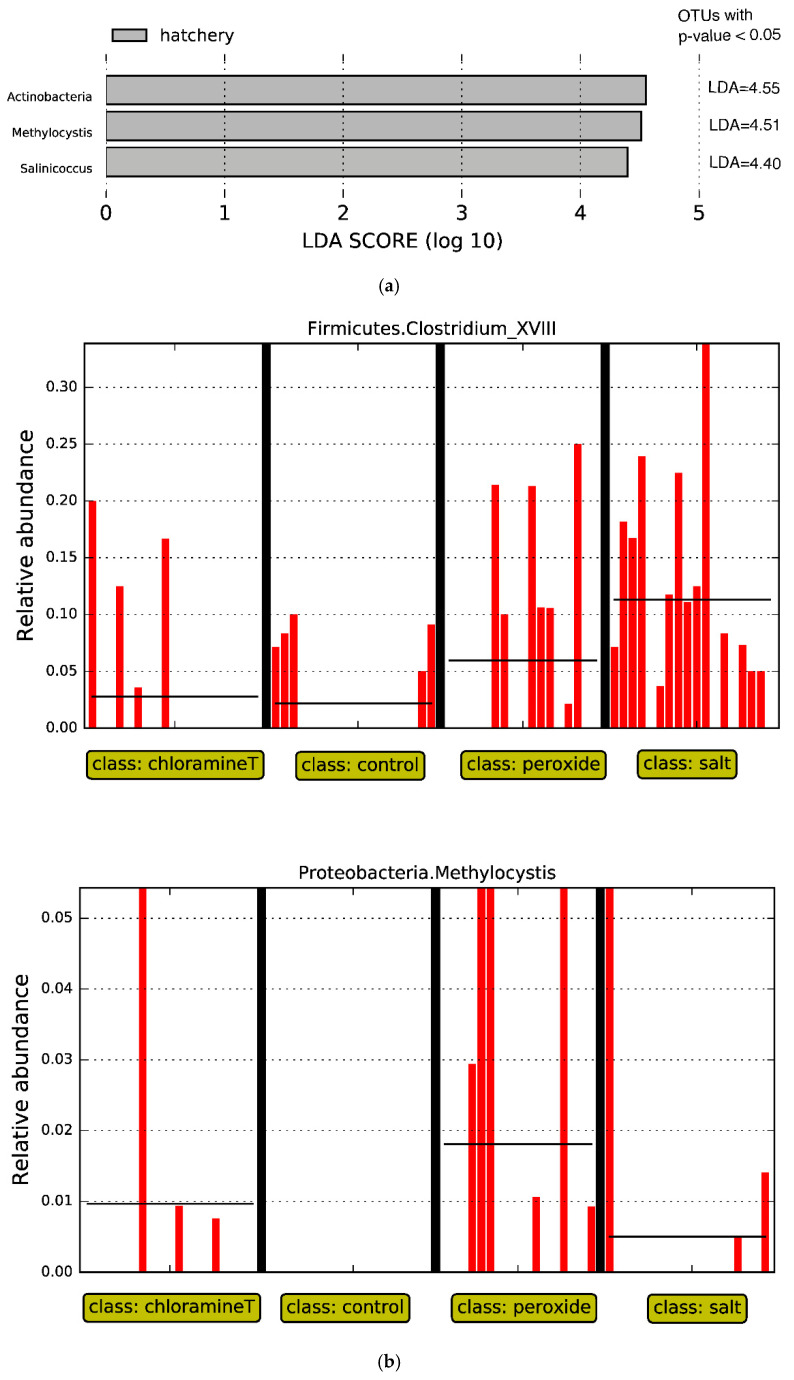
LEfSe analyses for (**a**) fish exposed to control treatment only, comparing hatchery and wild origins; (**b**) fish from all chemotherapeutant treatments against control across fish samples originated from wild; (**c**) fish from all chemotherapeutant treatments against control across fish samples originated from hatchery.

**Table 1 microorganisms-10-01005-t001:** PERMANOVA showing variability among fish gut microbiota across control groups only.

	Df	Sum Sq	Mean Sq	F-Model	R^2^	Pr (>F)
Origin (O)	1	0.416	0.416	1.792	0.050	0.107
Residuals	34	7.887	0.232		0.950	
Total	35	8.302			1.000	

**Table 2 microorganisms-10-01005-t002:** PERMANOVA showing variability among fish gut microbiota across all samples. Results revealed that origin effect (O) significantly influencing gut microbial communities composition for at least one sample across treatments and origins (PERMANOVA test permutation = 1000).

	Df	Sum Sq	Mean Sq	F-Model	R^2^	Pr (>F)
Treatment (T)	3	0.897	0.299	1.389	0.028	0.126
Origin (O)	1	1.699	1.699	7.900	0.052	*p* < 0.01
Residuals	139	29.900	0.215		0.920	
Total	143	32.496			1.000	

**Table 3 microorganisms-10-01005-t003:** PERMANOVA showing variability among fish gut microbiota across all samples originating from the hatchery.

	Df	Sum Sq	Mean Sq	F-Model	R^2^	Pr (>F)
Treatment (T)	3	1.108	0.369	1.913	0.076	0.012
Family (F)	1	0.402	0.402	2.084	0.028	0.057
Treatment (T) × Family (F)	3	0.737	0.246	1.272	0.050	0.191
Residuals	64	12.357	0.193		0.846	
Total	71	14.603		1.000		

**Table 4 microorganisms-10-01005-t004:** PERMANOVA showing variability among fish gut microbiota across all samples originated from the stream substrate (wild).

	Df	Sum Sq	Mean Sq	F-Model	R^2^	Pr (>F)
Treatment (T)	3	0.784	0.261	1.241	0.048	0.209
Family (F)	1	0.539	0.539	2.560	0.033	0.022
Treatment (T) × Family (F)	3	1.390	0.463	2.202	0.086	0.007
Residuals	64	13.471	0.210		0.832	
Total	71	16.184		1.000		

## Data Availability

Sequences have been deposited in the NCBI Sequence Reads Archive (SRA) under BioProject accession number PRJNA820564 (https://www.ncbi.nlm.nih.gov/sra/PRJNA820564, accessed 28 March 2022). The community matrix is provided in the Appendix A. The community matrix describing sequence counts for all OTUs for all treatments associated with this study can also be found on GitHub at https://github.com/ScribnerLab/Chemotherapeutants.git (doi.org/10.5281/zenodo.6418537, accessed on 22 April 2022).

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
