# Peer review of "Changes in Lake Sturgeon Gut Microbiomes Relative to Founding Origin and in Response to Chemotherapeutant Treatments"

_microorganisms, 2022, doi:10.3390/microorganisms10051005_

Round 1

Reviewer 1 Report

I reviewed the first version of this MS and the authors improved the quality of the MS and I suggest authors reading one more time to fix few language errors. Then, it would be ready for the final steps for acceptance.
Best regards

Author Response

Thank you

Reviewer 2 Report

The authors present data on sturgeon microbiome in response to treatment, which is absolutely necessary to maintain health. The manuscript is put together well, reads well, and contains a thorough analysis of a lot of data. The work is interesting in that it finds that founder effects are far more significant on the microbiota than treatment, which is useful for fisheries managers. As the samples were collected a decade ago, it will be interesting to see whether a follow up study sees similar components of the microbiome. Though arduous, longitudinal studies can be very informative. 

Author Response

Thank you for review

Reviewer 3 Report

The paper of Razak et al. reported on the possible changes in bacterial diversity that may be caused by biological features or chemical anthropic intervention.

The subject appears to be extremely interesting both from a pure and an applied research perspective. I particularly appreciated how the authors took advantage of a sound statistical approach to dissect the contributions of either factors mentioned above.

That said, I also have some comments and suggestions that, in my opinion would help increase the quality of the manuscript. At this stage, the text could be stremlined and both the concepts and the results could be presented in a more linear manner. This is especially true for the Results and Discussion sections. I'm not a native speaker myself, but a very different use of the English language was evident between the Abstract and rest of the manuscript. I'm not referring to typos (which are anyway present even though I'll not mark them), but to the construction and linearity of sentences.  

Please find below my comments, with corresponding line numbering:

-L76: add citation

-L80-95: this paragraph is unnecessarily detailed. The introduction holds even if this paragraph, which reads like a discussion, is removed

L102: authors could elaborate on this from an ecological perspective

L103-108: I suggest to move this paragraph before the one extending from L96-102

L113: reference 43 is 34 year old, bacterial communities were identified with the plaque technique. I suggest the authors to remove it in favour of a more recent one

L129-132: not clear. Why would SRFs pose such issues if few lines above you stated that they use natal water sources?

L134: define the term younf of the year. The readership of this Journal may not be familiar with this.

L142: it is unusual to state the overall working hypothesis by referencing a paper

L161-165: I'm not sure these lines fit in here

L167-172 and 194-197: here authors should just report on the methods employed. It is hoped that by line 167 the purpose of the study is understood by the reader.

L175: water temperature ranged between 19.9 and 26.3°C. Why did you transport on ice?

L184: please include an indication of the embryo developmental time prior to hatching for this species

L197: "naturally-produced fertilized eggs". Are you sure these are comparable in age (developmental time) to the hatchery-produced eggs.

L214: it would have been best to employ individual smaller tanks to surely exclude any contamination and, most importantly, perform weekly treatments without the need of moving fish.

Section 2.3: I have an important comment to make here. Why did the authors decide to filter natural river water with 50 um mesh? This clearance filters out particles that can be seen by human eye but certainly not bacteria. This flaw may compromise the overall purpose of the study, i.e. to clearly be able to dissect the effects of origin and treatments on bacterial diversity.

L230: why were fish rinsed with NON-sterile water following the treatments? Was this supposed to be a strategy to compensate for bacterial losses supposedly occurred because of the exposure to chemicals? The authors shoould clarify: 1) the reason why 50 um filter was used; 2) the rationale behind the rinsing with non-sterile river water; 3) whether river water was filtered fresh on a weekly basis; 4) whether, within each weekly treatment, river water was filtered separately for group 3, whose treatment lasted one extra day; 5) if the kinetic of bacterial community growth in non sterile river water was analyzed

Fig.1: given your multifactorial experimental design, this figure could be more informative

L267: the centrifugal force is missing

L279: report sample number per experimental group (plus biological and technical replicates)

L304: I don't think acronyms OTU and ASVs were expanded previously

L327: R packages musst be cited according to the correct reference

L330-333: these lines are not clear to me
section 2.6.2: please rewrite this paragraph, I had a hard time wrap my head around it. It could be written a much less confusing way. Also, this is where I note the most marked difference in the use of English between the abstract and this section. English must be improved.

L342-345: superfluous

L350-351: the focus of the analysis was already stated few lines above

L358: authors should clarify why they opted for testing separately for origin and then, for origin and treatment combined.

L356-365: was the exposure to peroxide of both hatchery and wild fish intended?

L400: I don't understand where these percentages values come from and where are they plotted -not in Fig. 2 I suppose.

L412-414: not clear.

L447: the figure legend says you have plotted standard errors, not confidence intervals.

Fig. 4: please report/declare/comment on the extremely low variability explained by all 4 models (0.033<r^2<0.076)

L488-491: isn't this implicit that beta diversity will not be significantly different if the Kruskal-Wallis test conducted on alpha diversity of control samples based on the origin factor also did not return significant results?

Table 2, building on comment on L358 (above): how do you explain that the origin factor is first non significant when tested alone and, instead, significant when tested in a model also considering the treatment factor?

L517-520: this is weird. The main test returned significant results, but the post-hoc did not. What is the significance in the first place due to?

L560-562: out of place.

L570: I suggest that the authors at least mention the possibility of a maternal control, which in fish models was demonstrated and reviewed to occur and regulate key biological process upon probiotic administration.

L578-579: this was already written and referenced in the introduction. Why repeating it?

L580-582: you shouldn't subtly imply that changes may occur if the treatment period were longer.

L592-593: the finding would indeed be important if the efficacy of the given prophylactic measures in fighting certain diseases were demonstrated, i.e. a functional test is lacking.

L596: the weekly periodicity aspect is suggested twice, here and at line 602.

L597-599: are the tested concentrations in line with real world usage?

L599-601: THIS is exactly what the lack of river water sterility (an issue I raised earlier) is relevant for.

L629-634: I don't see the link between your resultsand clinical studies. Also, you have not treated fish with antibiotics sensu strictu

L647: instead of trying to link your results to humans, why not try to address and reference the topic in other teleost species? The investigation of gut microbiota change in response to external stimuli has recently been reviewed in economically relevant mediterranean cultured fish species

L654-662: again, all references that have already been used in the introduction. With regards to ref. 46 (Liu et al), you could elaborate on this: from which perspective are results achieved by you and Liu et al. similar?

L670: please start the sentence by specifyng that those were control fish.

L671-673: this is interesting, as it suggest that microbiota colonizing the chorion may also penetrate it and colonize the developing GI tract.

L676-678: true, but don't forget you have washed them with NON-sterile river water.

L680-683: I don't see the relevance of this statement. In addition, you experimented with larval specimens, which are not competent immunity-wise at early ontogeny stages.

These are all the comments I had. Thanks for your attention.

Round 2

Reviewer 3 Report

Thank you for implementing some of the changes I suggested.